# The Effect of Green Finance on the Ecological and Environmental Quality of the Yangtze River Economic Belt

**DOI:** 10.3390/ijerph191912492

**Published:** 2022-09-30

**Authors:** Decai Tang, Hui Zhong, Jingyi Zhang, Yongguang Dai, Valentina Boamah

**Affiliations:** 1School of Management Science and Engineering, Nanjing University of Information Science & Technology, Nanjing 210044, China; 2School of Business, Nanjing Normal University, Nanjing 210023, China; 3Bache Middle School, Wujiang District, Suzhou 215000, China

**Keywords:** green finance, ecosystem quality, entropy method, panel data

## Abstract

Since China’s reform and opening up, the speed of economic development has increased significantly. However, at the same time, there are also serious environmental pollution problems. To resolve the deep-seated contradiction between economic growth and environmental protection, green finance has gradually gained attention in China’s development. Based on this, the paper explores the impact of green finance on the quality of the ecological environment in the Yangtze River Economic Belt. The main part of the paper is based on panel data of eleven provinces and cities in China’s 2011–2020 Yangtze River Economic Belt. Seven indicators, including chemical oxygen demand COD, harmless treatment rate of domestic waste, and green coverage rate of built-up, were used to construct an ecological and environmental quality evaluation index system. The entropy method is used to measure the ecological environment quality level and green finance development level of various provinces and cities in the Yangtze River Economic Belt. The impact of green finance development on ecological environment quality is analyzed using a panel data model. The research results show that: (1) The development level of green finance and the quality of the ecological environment in the Yangtze River Economic Belt have improved between 2011 and 2020. (2) The development of green finance has a significant positive impact on the quality of the ecological environment in the Yangtze River Economic Belt. In addition, related research has focused on the impact of green finance on a certain branch of ecological and environmental quality and lacks an analysis of the overall impact. Therefore, this paper constructs a comprehensive evaluation system for ecological environment quality and analyzes the overall impact of green finance on ecological environment quality in the region.

## 1. Introduction

Over the past 40 years of reform and opening up, China’s economic development has maintained a high rate of growth, and the country’s productivity and economic development have been steadily rising, its comprehensive national strength has continued to grow, and people’s income and quality of life have improved significantly. However, while China’s economy has been developing at a high rate, the development quality has not been satisfactory, with many non-desired by-products. Ecological and environmental problems are becoming increasingly serious, with air pollution, water pollution, and resource shortages becoming increasingly acute. Population, resource, and environmental issues are gradually becoming important factors limiting China’s sustainable development. China’s previous crude economic development model, characterized by high growth, high consumption, and low output, has become increasingly unsustainable. Green finance is gaining more and more attention in China as an innovative financial development model to resolve the deep-rooted conflict between economic growth and environmental pollution.

In the 1960s and 1970s, the industrial revolution made environmental problems in the economy and society increasingly prominent, and green finance also appeared in the public eye. Commercial banks mainly carried out the initial implementation of green finance and gradually expanded to non-bank financial institutions such as insurance companies and fund companies. Presently, some government agencies, enterprises, and other non-financial institutions have also joined the ranks of green finance. Although China’s green finance started late, it has shown a booming trend in recent years. In December 2015, the People’s Bank of China launched green financial bonds, officially launching China’s green bond market. In 2017, the 19th Party Congress proposed to promote green development, increase the protection of ecosystems and make efforts to solve environmental problems, to overcome ecological and environmental problems brought about by the previous development model. The 2019 government work report once again emphasized the need to accelerate the development of green finance, cultivate several environmental protection backbone enterprises and enhance green development capabilities. This all reflects the importance that the Chinese government attaches to the development of green finance.

Green finance has made great progress in China’s economic development. According to data from the China Banking and Insurance Regulatory Commission, the total amount of green credit from 21 major banks in China in 2020 has increased significantly compared to 2011; in addition, according to data shown in the China Environment Statistical Yearbook 2011–2020, the total investment in environmental pollution control in the Yangtze River Economic Zone has also increased from 243.85 billion yuan in 2011 to 471.95 billion yuan in 2020, an increase of about 1.94 times. This series of investments has been effective, and data from the China Environmental Statistics Yearbook shows that environmental pollution problems in China’s Yangtze River Economic Belt have been effectively curbed, with emissions of wastewater and general industrial solid waste in 2020 increasing by only about 1.16 and 1.05 times compared to 2011 and the rapidly developing economy.

It can be seen that the development of green finance in the Yangtze River Economic Zone has made great progress and has contributed significantly to the sustainable development of the local economy and society. Therefore, we can make a preliminary hypothesis from the data that green finance impacts the quality of the ecological environment

Although there is a great development of green finance in China’s Yangtze River Economic Belt presently, there are still environmental pollution problems. Therefore, this paper takes the green finance and ecological environment quality of the Yangtze River Economic Belt as the research object. Firstly, we measure and analyze the level of green finance development and ecological environment quality index in eleven provinces and cities in the Yangtze River Economic Belt by entropy value method to understand the changing trend of the two during 2011–2020. Then we use a panel data model to study the effect of green finance on eco-environmental quality and provide new ideas to achieve the coordinated development of economic growth and eco-environmental quality. Through a series of studies, this paper hopes to draw the attention of enterprises and investors to green finance. During the 13th Five-Year Plan period, Chinese cities will need 6.6 trillion yuan of green investment in building green transportation and clean energy alone, but the government can only provide 10% to 15%. Therefore, we need to encourage relevant companies to focus on ecological quality, understand the impact of green finance on the ecological environment, and invest more in green finance. This will help improve the utilization of energy resources for urban economic development, accelerate the development and utilization of new and renewable energy, and improve the ecological environment of cities.

In addition, previous academic studies have focused on the impact of green finance on the ecological environment of a country, such as China and Pakistan, and lacked research on a region or certain collection of regions. The Yangtze River Economic Belt is the most developed region in the entire Yangtze River Basin., covering 11 provinces and cities with huge development potential, including vast land resources, rich natural resources, huge population size and hidden domestic demand potential. Therefore, this paper selects this representative region to measure the level of green finance and the level of ecological environment quality in the Yangtze River Economic Belt and analyzes the impact of the former on the latter and regional differences. By considering the development of green finance in the Yangtze River Economic Zone, the role of green finance can be better utilized, and new ideas can be provided for the overall environmental governance in China.

In this paper, the literature and empirical research methods are used. The literature research method is based on a certain research purpose, through the investigation of literature, to obtain information to comprehensively and correctly understand the information related to the research problem. In this paper, the literature research method is used to understand the history and development status of green finance and determine the research topic of this paper. The empirical research method focuses on proposing a design and building a model based on existing theory and practice needs. In addition, through purposeful and systematic manipulation, the causal relationship between influencing factors and phenomena is determined by observing, recording, and measuring changes in the values of relevant phenomena. In order to generalize those factors that are not observed and do not vary over time but affect the explanatory variables, we chose a panel data model for the empirical study. Moreover, due to the need for a multi-indicator assessment system, we need to apply the entropy value model to assign weights to them.

## 2. Review of the Literature

### 2.1. The Importance of Green Financial Development

This paper reviews the research results on green finance at home and abroad and analyzes the relevant literature statistically. In terms of the number of papers published, there has been a significant increase in the number of green finance-related research papers published in the past five years. In terms of the research directions of the papers, they include various directions such as environmental science, business economics, engineering, and energy fuels. Therefore, combining the research results of domestic and foreign scholars, this paper mainly reviews the importance of green finance development.

Scholars at home and abroad have studied the important role of green finance from several perspectives. Firstly, in terms of sustainable development, Tolliver Clarence (2019) argued that green bonds, as an important part of green finance, are gaining prominence in climate change and sustainable development finance frameworks [1]. Sahar & Anis (2015) analyzed the adoption and application of the Equator Principles by 78 banks in 35 countries. They found that banks’ investment in green industries would create more market value and contribute to the sustainable development of the economy [2]. Furthermore, Nassani (2017) highlighted the importance of green policy tools linked to national policies for sustainable development [3]. While Iqbal, N (2020) argued that the government and policymakers should develop green financing policies to encourage environmental entrepreneurs to establish environmentally friendly businesses, promote environmentally friendly products use, reduce environmental problems and achieve sustainable development in Pakistan [4].

In terms of the impact on enterprises, Fakhar Shahzad (2020) argued that there is a significant positive correlation between green finance and corporate green performance [5]. Zhou (2019) argued that the issuance of green bonds positively impacts company share prices, profitability, and operating performance and helps to improve corporate social responsibility and value creation [6]. Wu (2021) found that with the rapid development of green finance in policy formulation and practice, there is a greater interest in enterprises engaged in environmental protection industry-related businesses [7]. Hu (2021) noted that green finance could improve the stock trading activities of firms through capital market effects and affect the long-term value of firms by improving their operational efficiency and profitability through practical effects [8]. Lai (2022) also argued that green credit significantly increases the value of new energy firms [9]. Cui (2021) emphasized that green finance is an important way to help firms achieve green transformation and development, guiding listed firms towards green development [10]. Peng (2021), on the other hand, argued that green credit policies significantly discourage debt financing for heavy polluters and encourage firms to eliminate backward production capacity [11]. Yang (2019) argued that green credit policies aim to reduce emissions of highly polluting enterprises by improving information disclosure in the lending process [12].

Regarding industrial structure transformation, some scholars also have their views. Gao (2022) believed green finance is necessary to promote industrial structure transformation and upgrading [13]. Li (2022) believed that the transformation and development of traditional finance to green finance had promoted the transformation of a traditional high-carbon economy to a low-carbon economy [14]. Chen (2021) pointed out that although with the help of green finance, the allocation of financial resources can be optimized and the transformation of industrial structure can be promoted, the overall efficiency of green finance in promoting industrial transformation and upgrading has shown a downward trend [15]. Shao (2021) believed that green credit could effectively guide the rational allocation of resources and promote the development of secondary and tertiary industries [16]. Xie (2022) argued that industrial structuring is associated with the level of green finance development through spillover effects [17]. Wang (2021) advocated that the government should actively play the role of green finance to promote industrial structure upgrading and technological innovation [18].

Regarding energy consumption and use of energy efficiency, scholars hold two main views, with one part arguing that green finance can improve energy use efficiency: Liu (2021) argued that green finance is a suitable and supportive financing tool for energy efficiency [19]. At the same time, Liu (2021) stated that green finance could promote renewable energy production and efficiency to achieve the desired results [20]. Peng (2021) argued that green finance could significantly improve energy efficiency and provide important support for developing policies to optimize the energy mix and improve energy efficiency [21]. A section argued that green finance could contribute to upgrading the energy mix: Wang (2021) argued that the development of green finance has contributed to the structural transformation of energy consumption from conventional to sustainable energy consumption [22]. Sun (2022), on the other hand, pointed out that advances in the green finance development index have slowed the growth of renewable energy use by reducing greenhouse gas emissions [23]. In addition, Lan (2021) argued that the promotion of green bonds facilitates bioenergy production and emission reduction [24]. Wang (2021) argued that introducing green credit promotes renewable energy investment and accelerates the energy transition [25]. Dogan, E (2022) investigated the connectedness and spillover relationship between green finance and renewable energy. The results show that dynamic connectedness, both total and pairwise, is heterogeneous over time and influenced by economic events [26].

In terms of the effect of green finance on green development, Zhang (2022) argued that green finance significantly contributes to green development efficiency when R&D investment is above a certain threshold [27]. Ye (2022), on the other hand, pointed out that there are regional differences in the impact of green finance on green development, with the impact in the eastern region being much greater than that in the central and western regions [28]. Liu (2020) argued that green finance had become a new growth point and core engine for promoting green development, with social responsibility and environmental benefits as the core of development [29]. Gianfrate and Peri (2019) suggested that green bonds are one of the best tools for mobilizing financial resources for green and sustainable investments [30].

### 2.2. Effects of Green Finance on the Quality of the Ecological Environment

Regarding green finance and eco-environmental quality, some scholars focus more on the direct effect between green finance and eco-environmental quality. For example, Huang (2021) concluded that green finance and environmental quality have a significant positive spatial autocorrelation [31]. At the same time, Li (2021) built on this, concluding that the development of green finance promotes the improvement of the local ecological environment and found a significant positive spatial spillover effect [32]. Zeng (2022) argued that green finance has a significant negative impact on urban haze pollution [33], thus reducing the ecological quality. While Huang (2021), confirming the effect between the two, proposed the establishment of green finance pilot zones to reduce environmental pollution and improve the environment through green finance policies [34]. Zhou (2020) further proposed that green finance not only has a positive effect on environmental improvement, but the improvement effect varies depending on the level of economic development [35]. Cai (2021) argued that green finance is a new financial model that can protect the environment and improve economic efficiency [36].

In addition, carbon emissions are an important factor affecting the quality of the ecological environment, and green finance can also affect the quality of the environment by influencing carbon emissions. Hence, some scholars emphasize the role of green finance on carbon emission reduction: Ren (2020) argued that increasing China’s green finance development index is beneficial for reducing carbon intensity [37]. Wang (2021) argued that green finance instruments have a significant negative impact on carbon emissions intensity and can adapt to environmental regulations of different intensities and synergistically promote carbon reduction [38]. Guo (2022) argued that green finance can significantly reduce agricultural carbon emissions [39]. Wang (2021) argued that green finance is the best financial strategy to reduce carbon dioxide emissions [40]. Hu (2022) argued that green credit could reduce carbon emissions through energy mix and intensity [41]. Zhang (2021) argued that green finance is a powerful measure to promote global carbon emission reduction [42]. Yu (2021) argued that green finance is a financing method that promotes the development of environmental industries by reducing greenhouse gas emissions [43]. Hao (2021) argued that the green credit policy is one of the main policies that can help reduce sulfur dioxide emissions [44]. Sun (2021) suggested that corporate carbon accounting can also contribute to reverse regional green finance development [45]. Glomsrød and Wei (2018) predict that if green bonds develop normally, avoided output could reach the same CO2 emissions as the EU and Japan combined in almost a year by 2030 [46].

### 2.3. Comprehensive Analysis of the Literature

From the comprehensive analysis of the above literature, we can find that domestic and international literature mainly focuses on the relationship between green finance and sustainable development, the relationship between green finance and corporate value, industrial structure transformation, energy use efficiency, and green development. However, there are some differences between domestic and foreign scholars’ research content on the effect of green finance on the ecological environment. Chinese scholars mainly focus on the direct impact effect of green finance on ecological and environmental quality or through carbon emissions. For example, Li (2022) considered green finance an important driver to achieving carbon neutrality and coordinating economic development and environmental governance [47]. In contrast, Kong (2022) argued that green finance development promotes the advancement of new energy technologies, thus helping to achieve carbon neutrality goals, reduce environmental pollution, and promote an ecological environment [48]. Foreign scholars, on the other hand, study the impact of green finance on the quality of the ecological environment through other transmission mechanisms. For example, Poberezhna (2018) investigated the advantages of the green economy and blockchain and the possibility of combining both of them and proposes that such a combination addresses global water scarcity and thus reduces the threat of environmental degradation [49]. Romano (2017), on the other hand, suggested that an active green finance policy issued by the government may increase investments in the renewable energy sector, provide environmentally friendly businesses providing policy support, and thus improve environmental quality [50]. MacAskill, S argued that green bonds are emerging as an influential financing mechanism for climate change mitigation and suggests that bond pricing should take into account the environmental preferences of investors so that green bonds become a catalyst for mitigating global climate change and thus slowing down environmental degradation [51]. The reason for this difference is that at this stage, the green financial system in Europe, the United States, and other developed countries is more mature, and relevant scholars have studied the transmission mechanism of green finance to improve the ecological environment more perfectly. In contrast, the development of green finance in China started late and was set up by national government agencies from the top down. Green finance standards have not been unified regarding project evaluation and certification.

In the above studies, there are certain research gaps, such as the research mainly focusing on the effect of green finance on the ecological and environmental quality of a country, a province, or city, and lacks research on a certain economic region. In addition, different scholars choose different influencing factors in constructing a comprehensive evaluation system, so the results obtained have certain errors. This paper takes the green finance and ecological environment quality of the Yangtze River Economic Zone as the main research object, constructs a comprehensive evaluation system of green finance and ecological environment quality with reference to the existing studies of scholars, and analyzes the relationship between the two using a panel regression model. This study helps to analyze the impact of green finance on the ecological environment quality of the Yangtze River Economic Belt, improve the theory of green finance, and provide new ideas to realize the coordinated development of green finance and ecological environment quality.

## 3. Research Methodology

### 3.1. Literature Review Method

The literature review of this paper consists of three main parts: a review on the independent variable green finance, a review of the relationship between the level of development of green finance and the quality of the ecological environment, and a summary.

In this paper, regarding the review of independent variables, only the main variable green finance, is considered. The main content is a study on the importance of green finance, includes the effect of green finance on sustainable development, enterprise development, industrial structural transformation, energy consumption, energy use efficiency, and green development. The above is mainly in the first part of the literature review. The second part focuses on the academic literature on the relationship between the level of green finance development and the quality of the ecological environment. This section is the most similar to the research theme of this paper, from which we have identified gaps in the literature. We, therefore, conclude in the third part of the literature review with the research theme of this paper: a study on the effect of green finance on the ecological and environmental quality of the Yangtze River Economic Belt.

### 3.2. Entropy Value Method

#### 3.2.1. Data Standardization

In this paper, a panel data model was selected for research before the empirical study. Due to the needs of the multi-indicator evaluation system, the entropy method needs to be applied to assign weights to them. Before using the entropy method to calculate the weights of each indicator, we had to normalize the indicators. This is because in a multi-indicator assessment system, due to the differences in the nature of the indicators, their metric scales and quantitative levels are different. If raw data is used for the analysis, it will highlight the importance of data with high values in the comprehensive analysis. So, the main purpose of this step is to eliminate the quantitative relationships between variables so that the data fall into a smaller specific interval and that indicators of different nature can be compared and weighted. First, we select m indicators for a total of n samples, then $X_{\mathrm{ij}}$ is the value of the jth indicator of the ith sample, i = 1, 2, 3, … n; j = 1, 2, 3, … m.

We then normalized the data using the following equations.
(1)$$x_{\mathrm{ij}}=\frac{X_{\mathrm{ij}}-\min\left\{X_{j}\right\}}{\max\left\{X_{j}\right\}-\min\left\{X_{j}\right\}}$$
(2)$$x_{\mathrm{ij}}=\frac{\max\left\{X_{j}\right\}-X_{\mathrm{ij}}}{\max\left\{X_{j}\right\}-\min\left\{X_{j}\right\}}$$

If the change in the impact of the indicator on the quality of the ecosystem is positive, the calculation of Equation (1) is used. If the change in the impact of the indicator on the quality of the ecosystem is negative, the calculation of Equation (2) is used. $X_{\mathrm{ij}}$ represents the standardized values, max{$X_{j}$} and min{$X_{j}$} denotes the maximum and minimum values of each indicator in all years, respectively.

#### 3.2.2. Calculating Sample Weights

It is mainly to calculate the weight of the ith sample under the jth indicator for that indicator
$$P_{\mathrm{ij}}=\frac{X_{\mathrm{ij}}}{\sum_{i=1}^{n}X_{\mathrm{ij}}}$$

#### 3.2.3. Calculate the Entropy Value of the jth Indicator

Primarily, the entropy value of the jth indicator is calculated.
$$e_{j}=-K*\sum_{i=1}^{n}(P_{\mathrm{ij}}*\ln(P_{\mathrm{ij}}))$$

#### 3.2.4. Calculate the Coefficient of Variation for Indicator j

The information utility value of an indicator depends on the difference between the information entropy of the indicator and 1. The higher the information utility value, the greater the importance of the evaluation and the greater the weighting.
$$d_{j}=1-e_{j}$$

#### 3.2.5. Calculation of Evaluation Indicator Weights

The main purpose is to calculate the weight share of the jth indicator in the overall evaluation system.
$$\omega_{j}=\frac{d_{j}}{\sum_{j=1}^{m}d_{j}}$$

#### 3.2.6. Calculating the Sample Composite Index

The final composite index of the sample, in this article, refers to the Green Finance Development Index and the Eco-environmental Quality Index.
$$z_{i}=\overset{m}{\underset{j=1}{\sum }}\omega_{j}X_{\mathrm{ij}}$$

## 4. Empirical Studies

### 4.1. Selection of the Sample and Data Sources

#### 4.1.1. Explanatory Variable: Green Finance Development Index (GF)

Green finance development level (GF): China’s green finance is mainly influenced by the data on green investment, green funds, green credit, green securities, and carbon finance but is affected by the availability and scientific nature of the data. This paper mainly takes the two major data of green investment and green credit to measure the green finance development level of China’s Yangtze River Economic Zone. Green investment is mainly based on the proportion of investment in environmental protection in the GDP of each province and city in the Yangtze River Economic Zone as a proxy indicator. Its sample data is mainly obtained from the China Environmental Statistics Yearbook. Green credit is mainly based on the share of interest expenditure of high energy-consuming industries in the total industrial interest expenditure of each province and city in the Yangtze River Economic Zone as its inverse indicator, with its main data coming from the China Statistical Yearbook and the China Industrial Statistical Yearbook.

In assigning weights to the proxies, we chose the most commonly used entropy method to calculate the weights for both green credit and green investment data, and the results are shown in Table 1. The information entropy value e represents the average amount of information after redundancy has been excluded from the information. The size of the information utility value d determines the size of the weight, which represents the importance of a particular indicator to the evaluation.

From the table above, we can see that the weights of green credit and green investment are 0.333 and 0.667, respectively, and the weights are relatively evenly distributed between the two items, both around 0.5. We can then derive the following formula for measuring green finance.
$$\mathrm{GF}=0.33+0.67X_{2}$$

$X_{1}$ represents green credit and $\;X_{2}$ represents the green investment. According to the formula, we derived the green financial development index for the eleven provinces and cities in the Yangtze River Economic Belt from 2011–2020, as shown in Table 2.

We can see from the table the specific values of the level of green finance development in each province and city in the Yangtze River Economic Belt, with Anhui, Jiangxi, Zhejiang, Jiangsu, Chongqing, and Shanghai having a high level of green finance development, and several provinces in Anhui, Hunan, and Guizhou having a more fluctuating level of development. The overall green finance development index of the Yangtze River Economic Belt peaked in 2013, and although it has decreased in recent years, it is generally relatively stable.

#### 4.1.2. Explanatory Variable: Composite Index of Ecological and Environmental Quality (E)

When measuring the level of ecological and environmental quality of the Yangtze River Economic Belt, due to the complexity of the influencing factors, we therefore measure and evaluate the ecological and environmental quality of the Yangtze River Economic Belt in four dimensions: environmental pollution, environmental management, environmental construction, and energy consumption objectively and scientifically. The comprehensive evaluation system is constructed, as shown in Table 3.

In the dimension of environmental pollution, three indicators are selected: COD, SO_2_, and general industrial waste emissions. In comparison, the first two indicators represent wastewater emissions and waste gas emissions in each region, respectively. All three indicators negatively impact the quality of the ecological environment. In the dimension of environmental governance, a single indicator, the rate of harmless disposal of domestic waste, was selected and the impact of this indicator on the quality of the ecological environment was positive. In the dimension of environmental construction, two indicators, namely the greening coverage rate of built-up areas and the area of parkland per capita, were selected to reflect the level of environmental protection and greening in each region. These two indicators’ impact on the ecological environment’s quality was also positive. For the dimension of energy consumption, the total energy consumption was chosen, which negatively impacts the quality of the ecological environment.

The weights of the above data were measured by the entropy method, and the measured results are shown in Table 4.

The entropy method was used to calculate the weighting of a total of seven indicators, including chemical oxygen demand, and the table above shows that the weighting coefficients for the seven indicators are 20.53%, 15.48%, 17.01%, 4.97%, 10.44%, 21.06%, and 10.52%, respectively. There are some differences in the weighting of each item, with the highest weighting of 21.06% for the parkland area per capita and the lowest weighting of 4.97% for the household waste disposal rate. We then multiplied the standardized index data with the corresponding weighting coefficients to obtain the scores of individual indicators in the comprehensive evaluation system. Furthermore, we derived the comprehensive index of ecological and environmental quality in Table 5.

In order to visualize the trend of ecological environment quality in 11 provinces and cities in the Yangtze River Economic Belt, Figure 1 is shown below:

As can be seen from Table 5, the ecological environmental quality of all provinces and cities in the Yangtze River Economic Belt fluctuated to varying degrees during the period 2011–2020. Generally, it showed an upward trend, with the ecological environmental quality of Jiangsu, Zhejiang, Jiangxi, Hubei, Hunan, and Chongqing all reaching their highest values in 2018, and the high points of the ecological environmental quality of the remaining provinces and cities mostly in 2016 and 2017. Although the indicators fell back in 2019 for all provinces and cities, the overall ecological and environmental quality improved over 2011. There are also regional differences between different provinces and municipalities, with Chongqing, Shanghai, Jiangxi, Zhejiang, Guizhou, and Anhui having relatively better ecological and environmental quality. This reflects the differences in regional government investment in the environment in each region and is also influenced by each province and city’s own geographical location and degree of economic development.

#### 4.1.3. Control Variables

a.Industry structure

Industrial structure refers to the production links and proportional relationships between the various production sectors of the national economy, and there are three main types of industries in China. When the primary, secondary, and tertiary industries account for 5%, 25%, and 70% of the national economy, respectively, the economic structure tends to be reasonable, and the operation of the whole country’s economic system will be an ideal state [52]. In order to achieve this ideal state, we need to upgrade the industrial structure actively. With the continuous optimization and adjustment of China’s industrial structure, resources have been optimally allocated and gradually transferred to environment-friendly industries, reducing ecological and environmental pollution and promoting ecological and environmental quality improvement. Therefore, this paper selects industrial structure as the control variable and the proportion of value added in the tertiary industry to GDP as a proxy indicator.

b.Level of economic development

The level of economic development is one of the important factors affecting the quality of the ecological environment. As the economy develops, it will inevitably increase waste emissions, exacerbate environmental pollution, and affect the quality of the ecological environment. Still, economic development will also lead to the development of green technology and reduce pollutant emissions from highly polluting enterprises through technology. Therefore, this paper selects the level of economic development as the control variable and the real per capita GDP of each province and city as a proxy indicator.

c.Level of urbanization

The higher the level of urbanization, the more resources humans consume daily and the lower the land cover, contributing to the deterioration of the ecological environment. Therefore, this paper selects the level of urbanization as the control variable and uses the proportion of the urban population to the total population in each province and city as a proxy indicator.

#### 4.1.4. Summary and Description of Variables

We have selected explanatory variables, explained variables, control variables and explained the reasons for the selection of the above variables in the previous subsections. This section focuses on a brief summary of all the variables and the meaning of the variables, as shown in Table 6.

### 4.2. Descriptive Statistics of Variables

The descriptive statistics for the panel data analysis consisting of the variables in the previous section are shown in Table 7.

As can be seen from Table 7, the mean value of the explained variable ecological environment quality index is 0.603, the standard deviation is 0.127, the maximum value is 0.907, and the minimum value is 0.361. The ecological environment quality level of the provinces and cities in the Yangtze River Economic Zone is within a certain range, and the data is relatively stable. The mean value of the explanatory variable Green Financial Development Index is 0.435, the standard deviation is 0.161, the maximum value is 0.94, and the minimum value is 0.08; the data is also relatively stable. From the statistics of the control variables, the standard deviation of the two variables of industrial structure level and urbanization level is small. The difference between the maximum and minimum values is small. The data is stable, indicating that the regional differences in the proportion of the added value of the tertiary industry to GDP and urbanization level among the eleven provinces and cities in the Yangtze River Economic Belt are not significant. In contrast, the standard deviation of the variable of GDP per capita is large, and the difference between the maximum and minimum values is large, indicating that the regions’ differences are large.

### 4.3. Model Building

This study uses the level of green financial development as the explanatory variable, GDP per capita, value added of the tertiary industry as a proportion of GDP, the urban population as control variables, and the ecological environment quality index as the explained variable to construct a panel model as follows.
(3)$${\mathrm{EQ}}_{i,t}=\beta_{0}+\beta_{1}{\mathrm{GF}}_{i,t}+\beta_{2}{\mathrm{IS}}_{i,t}+\beta_{3}{\mathrm{RPGDP}}_{i,t}+\beta_{4}{\mathrm{UL}}_{i,t}+\mu_{i,t}$$

In the above equation, EQ stands for Eco-environmental Quality Index, GF stands for Green Financial Development Index, IS stands for Industrial Structure, RPGDP stands for Economic Development Level, UL stands for Urbanization Level, i stands for province, t stands for year, $\beta_{0}$ stands for intercept term, $\mu_{i,t}$ stands for the random error term, ${\;\beta }_{i}$ stands for regression coefficients of each variable.

The next step is to perform a model test to find the optimal model; the results are shown in Table 8.

From the above table, we can see that the F-test is significant at the 5% level, implying that the fixed-effects model is more suitable than the mixed-effects model, and the BP-test is significant at the 5% level, implying that the random-effects model is also more suitable than the mixed-effects model. The Hausman test shows a significant level of 5%, implying that the fixed effects model is better, so we chose the fixed effects model.

### 4.4. Empirical Analysis

After selecting the appropriate model, further empirical analyses were conducted, and the final results are shown in Table 9.

The regression analysis was carried out with the level of green finance, the value added of the tertiary sector as a proportion of GDP, the proportion of the urban population as independent variables, and the ecological environment quality index as the dependent variable.

The final analysis shows that the level of green financial development is significant at 0.01 (t = 2.916, *p* = 0.004 < 0.01), and the regression coefficient value is 0.173 > 0. This indicates that the level of green financial development in the Yangtze River Economic Zone has a significant positive influence on the region’s ecological and environmental quality indicators. The regression coefficient is 0.011 > 0, indicating that the urban population ratio has a significant positive influence on the eco-environmental quality index. The regression coefficient is 0.011 > 0, indicating that the proportion of the urban population positively affects the EQI.

## 5. Conclusions

Based on panel data from eleven provinces and cities in the Yangtze River Economic Belt, this paper investigates the effect of the level of green financial development on the level of ecological and environmental quality and draws the following conclusions.

(1) In terms of the level of green finance, the green finance index passed the 5% significance test with a regression coefficient value of 0.173, which means that the level of green finance development will have a significant positive impact relationship on the ecological and environmental quality index. We should therefore promote the development of green finance through the government guiding social capital into green industries, optimizing the allocation of green financial resources, reducing capital investment in high pollution and high emission enterprises, and tilting social capital towards green industries that are environmentally friendly and energy efficient, promoting the reuse of renewable energy, and reducing carbon emissions from polluting enterprises.

(2) Regarding the industrial structure, the share of tertiary industry value added in GDP fails the 5% significance test, which means that the share of tertiary industry value added in GDP and the eco-environmental quality index will not have a significant relationship. China’s financial institutions have been aiming to improve the efficiency of capital allocation by increasing investment in new industries and reducing investment in highly polluting industries. Although the model regression results are not significant, the Chinese government and enterprises still need to guide the transformation and optimization of industrial structure [53], accelerate industrial restructuring [54], and promote the transfer of labor and capital from the primary and secondary industries to the tertiary industry.

(3) In terms of economic development, on the one hand, China’s economic development inevitably produces industrial wastewater, industrial waste gas, and general solid waste, the production of which affects the quality of the ecological environment. However, at the same time, economic development will also accelerate the development of science and technology, improve people’s living standards and increase investment in the living environment. Although the model regression results are not significant, in the long run, promoting economic development is conducive to improving the ecological environment.

(4) Regarding urbanization level, the proportion of the urban population passes the 5% significance test, implying that it has a significant relationship with the EQI. The level of urbanization is influenced by multiple factors, such as the level of economic development, technological innovation, and the size of the city [55]. As the size of cities increases, the population of cities and towns increases, and the quality of residents also improves. People seek a higher quality of life and a better production environment, which drives the government to pay more attention to the ecological quality of cities and towns and therefore has a significant role in the ecological quality of the environment.

In terms of the content of the study, the acquisition of data, and the selection of indicators, there are still shortcomings in this study, which are again listed in the hope of exploring the direction for future research.

The level of green financial development and eco-environmental quality of the Yangtze River Economic Zone is a complex and large system, including economic systems, ecosystems, etc. Therefore, this study cannot analyze and discuss the level of green development and eco-environmental quality of the Yangtze River Economic Zone comprehensively and thoroughly.As the development of green funds and green securities in China is not yet complete, and there are limitations in obtaining some statistics on green credit when measuring the level of green development in the 11 provinces of the Yangtze River Economic Belt, only data on green investment and green credit in each province and municipality from 2011 to 2020 were selected, and long-term indicators were not obtained. It is also difficult to obtain detailed statistics on the specifics of the municipal units in each province of the Yangtze River Economic Belt, making it difficult to conduct a more detailed analysis.In this paper, all factors influencing the level of ecological and environmental quality in the Yangtze River Economic Zone have been selected as far as possible in measuring ecological and environmental quality levels. A comprehensive and adequate integrated index system of ecological and environmental quality has been established as far as possible. Still, as the specific indicators for measuring each subsystem are determined considering the ease of data availability, the evaluation results of this paper are relatively rough.

## 6. Suggestions

As a result of the above model regression results, we can conclude that green finance positively impacts the ecological environment quality of the Yangtze River Economic Zone. Therefore, the suggestions below are made to promote the development of green finance in a high-quality manner, thereby promoting the improvement of ecological environment quality.

(1) The government should focus on supporting green energy, green consumption, and green investment. Firstly, enterprises should be encouraged to use green energy, and the selection of enterprises using green energy should be carried out regularly. This will contribute to the growth of new green industries and high-quality economic development [56]. In addition, consumers should be encouraged to make green consumption. People should be made aware of green energy through regular environmental public welfare activities to encourage them to use less or no disposable consumer goods and to promote renewable energy use. Finally, the government should encourage more social capital to invest in environmental protection, increase green investment, and guide social capital into green industries through a series of policies.

(2) China’s green financial products mainly include green credit, green bonds, green funds, carbon financial products, etc. Among them, the first two take up the heaviest proportion. Still, most of China’s existing green financial products have the disadvantage of being small in scale, making it difficult to support the number of funds required for green projects. For some projects with a long cycle, it is also difficult to match the duration of green credit. Green financial projects can effectively promote regional green technological innovation [57] and promote green financial development, so to further support the development of green finance and encourage the research and development of innovative green technologies [58], we need to promote more long-term investment development, such as green equity investment, to finance the operation of green projects better. We also need to promote innovation in how green credit is secured and collateralized and increase the proportion of green credit in total loans to promote the development of green finance.

(3) China should build green financial market financing and institutional construction based on the current international development trend of a low-carbon economy. Firstly, the government should set up objective and scientific green financial evaluation systems for enterprises involved in environmental protection, resource conservation, and clean energy, and give policies and resources to the corresponding enterprises, such as adjusting rediscounting and refinancing policies to expand the green financial business. Secondly, disclosure standards should be unified for financial products such as green credit and green investment to avoid unnecessary losses to investors caused by different disclosure standards. In addition, the green financial regulatory system should be improved. China’s current regulation of the financial industry mainly adopts a model of sectoral regulation, and for the regulation of green finance, the relevant operating rules can be improved.

## Figures and Tables

**Figure 1 ijerph-19-12492-f001:**
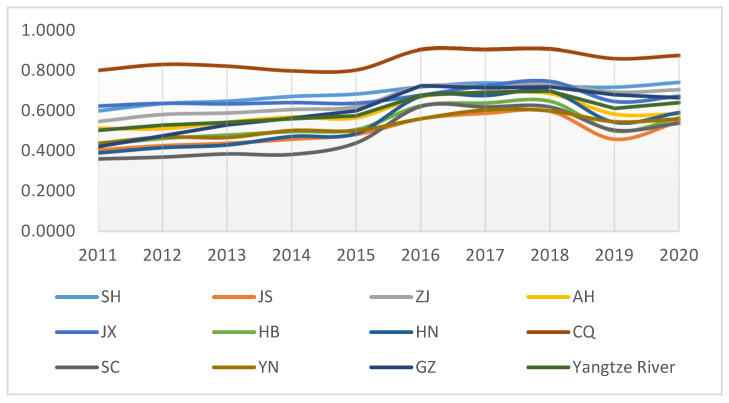
Ecological and environmental quality of 11 provinces and cities in the Yangtze River Economic Belt.

**Table 1 ijerph-19-12492-t001:** Summary of the results of the entropy method for calculating weights.

Item	Information Entropy Value e	Information Utility Value d	Weighting Factor ω
Green Credit	0.9897	0.0103	33.34%
Green Investment	0.9795	0.0205	66.66%

**Table 2 ijerph-19-12492-t002:** Green Financial Development Index of 11 provinces and cities in the Yangtze River Economic Belt.

	2011	2012	2013	2014	2015	2016	2017	2018	2019	2020
SH	0.4031	0.4038	0.4685	0.5038	0.5420	0.4321	0.3771	0.3969	0.3586	0.3069
JS	0.5151	0.5304	0.6147	0.5849	0.5941	0.4912	0.4514	0.3829	0.3552	0.3784
ZJ	0.4689	0.5660	0.5533	0.5410	0.5806	0.6329	0.4906	0.4188	0.3936	0.4334
AH	0.6651	0.7340	0.9411	0.7737	0.8041	0.8104	0.7509	0.5684	0.4676	0.4239
JX	0.6264	0.7253	0.5345	0.4662	0.5149	0.5976	0.5554	0.5493	0.5984	0.5027
HB	0.4228	0.4356	0.3917	0.3466	0.4448	0.5377	0.4838	0.3520	0.3811	0.3193
HN	0.2668	0.3567	0.3779	0.6563	0.3637	0.3293	0.3426	0.2728	0.2513	0.2836
CQ	0.8686	0.5681	0.5137	0.3713	0.4709	0.3861	0.4860	0.4001	0.3664	0.4102
SC	0.2883	0.3294	0.3634	0.2655	0.3745	0.3352	0.3263	0.3291	0.3043	0.3287
YN	0.3484	0.3076	0.4148	0.2234	0.2569	0.2090	0.1790	0.1364	0.1034	0.0788
GZ	0.2840	0.2568	0.3759	0.3663	0.5074	0.2890	0.4483	0.4450	0.2812	0.4128
Yangtze River Economic Belt	0.4689	0.4740	0.5045	0.4636	0.4958	0.4592	0.4447	0.3865	0.3510	0.3526

**Table 3 ijerph-19-12492-t003:** Construction of a comprehensive evaluation system.

Comprehensive Indicators	Guideline Level	Proxy Indicators	Properties
Eco-environmental Quality Index	Environmental pollution	COD (Chemical Oxygen Demand)	Negative
		Sulfur dioxide emissions	Negative
		General industrial solid waste emissions	Negative
	Environmental Governance	Harmless disposal rate of domestic waste	Positive
	Environmental construction	Greenery coverage in built-up areas	Positive
		Green space per capita	Positive
	Energy consumption	Total energy consumption	Negative

**Table 4 ijerph-19-12492-t004:** Summary of the results of the entropy method for calculating weights.

Item	Information Entropy Value e	Information Utility Value d	Weighting Factor ω
COD (Chemical Oxygen Demand)	0.9730	0.0270	20.53%
Sulphur dioxide emissions	0.9796	0.0204	15.48%
General industrial solid waste emissions	0.9776	0.0224	17.01%
Harmless disposal rate of domestic waste	0.9935	0.0065	4.97%
Greenery coverage in built-up areas	0.9863	0.0137	10.44%
Green space per capita	0.9723	0.0277	21.06%
Total energy consumption	0.9861	0.0139	10.52%

**Table 5 ijerph-19-12492-t005:** Eco-environmental quality index.

	2011	2012	2013	2014	2015	2016	2017	2018	2019	2020
SH	0.5999	0.6359	0.6486	0.6718	0.6841	0.7194	0.7390	0.7238	0.7180	0.7421
JS	0.4053	0.4268	0.4385	0.4587	0.4828	0.5606	0.5879	0.5978	0.4584	0.5471
ZJ	0.5469	0.5820	0.5894	0.6068	0.6207	0.7218	0.7219	0.7339	0.6935	0.7056
AH	0.5134	0.5112	0.5436	0.5691	0.5658	0.6704	0.6919	0.6860	0.5839	0.5854
JX	0.6242	0.6375	0.6346	0.6413	0.6383	0.6781	0.7252	0.7471	0.6468	0.6730
HB	0.4304	0.4624	0.4796	0.4968	0.5070	0.6257	0.6391	0.6479	0.4998	0.5646
HN	0.3904	0.4169	0.4303	0.4726	0.4874	0.6730	0.6762	0.6981	0.5431	0.5917
CQ	0.8016	0.8311	0.8225	0.7989	0.8034	0.9043	0.9050	0.9073	0.8602	0.8760
SC	0.3608	0.3699	0.3857	0.3832	0.4408	0.6224	0.6197	0.6194	0.5041	0.5388
YN	0.4384	0.4693	0.4673	0.5023	0.5017	0.5611	0.6044	0.6002	0.5462	0.5600
GZ	0.4222	0.4758	0.5299	0.5640	0.6005	0.7254	0.7147	0.7196	0.6800	0.6655
Yangtze River Economic Belt	0.5030	0.5290	0.5427	0.5605	0.5757	0.6784	0.6932	0.6983	0.6122	0.6409

**Table 6 ijerph-19-12492-t006:** Summary and description of variables.

Variable Type	Variable Name	Meaning of Variables	Variable Abbreviations
Explanatory variables	Green Finance Development Index	Green Investment and Green Credit	GF
Explained variables	Eco-environmental quality composite index	Level of ecological quality	EQ
Control variables	Industrial structure	Tertiary sector value added as a proportion of GDP	IS
	Level of economic development	Real GDP per capita	RPGDP
	Level of urbanization	Urban population as a proportion of total population	UL

**Table 7 ijerph-19-12492-t007:** Basic indicators.

Designation	Sample Size	Minimum Value	Maximum Value	Average	Standard Deviation	Media
Eco-environmental Quality Index	110	0.361	0.907	0.603	0.127	0.600
Green Finance Development Index	110	0.080	0.940	0.435	0.161	0.405
Tertiary sector value added as a proportion of GDP	109	0.330	0.730	0.472	0.086	0.470
Percentage of population in urban areas	110	35.030	89.600	57.974	13.397	55.715
GDP per capita	110	16,413	157,279	57,626	30,310	47,897

**Table 8 ijerph-19-12492-t008:** Summary of test results (n = 109).

Type of Test	Purpose of the Test	Test Value	Test Conclusion
F-test	FE model and POOL model comparison selection	*F* (10,94) = 37.832, *p* = 0.000	FE Model
BP test	RE model and POOL model comparison selection	χ^2^(1) = 158.775, *p* = 0.000	RE Model
Hausman test	FE model and RE model comparison selection	χ^2^(4) = 17.948, *p* = 0.001	FE model

**Table 9 ijerph-19-12492-t009:** Summary of panel regression models.

Item	FE Model
intercept distance	−0.218 (0.244)
Level of Green Financial Development	0.173 *** (2.916)
GDP per capita	0.000 (1.358)
Tertiary sector value added as a proportion of GDP	0.093 (0.453)
Percentage of population in urban areas	0.011 **** (5.339)
*R* ^2^	0.858
*R*^2^ (within)	0.579
Sample size	109
Testing	*F* (4,94) = 32.376, *p* = 0.000
Dependent variable: Eco-environmental quality index	

*** *p* < 0.01 **** *p* < 0.001.

## Data Availability

Not applicable.

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
