# Peer review of "The Effect of Green Finance on the Ecological and Environmental Quality of the Yangtze River Economic Belt"

_ijerph, 2022, doi:10.3390/ijerph191912492_

Round 1

Reviewer 1 Report

The article is executed well, the topic is timely, and the paper can contribute to the current academic debate.

Nevertheless, the author might improve the introduction section. They might explain better the details of their study, the method, the motivation, the contribution, the results and so on.

the reader should be distracted because the authors developed deeper aspects that should be developed in other sections. The authors might summarize the introduction with highlights on background, topic, the gap in the literature, contribution and significance, to explain in detail in the theoretical background section.

I would again suggest the same for the methodology section for greater clarity.

Good luck!!!

Author Response

Dear reviewer, thank you very much for your valuable comments and suggestions. According to your proposal, we have revised the paper, to get your approval.

Point 1: The introduction section needs to be improved to better explain the details of the study, the methodology, the motivation, the contributions, the results, etc.

Response 1: Thank you for your suggestion. We have improved the introduction section by focusing on the study’s background, motivation, significance, and contribution. In the other sections, more in-depth aspects are presented. Please see Line 114-137.

Point 2: The authors might summarize the methodology section and explain in detail in the theoretical background section.

Response 2: We briefly describe the research methodology of this paper in the preceding sections and further elaborate on it in the subsequent chapters. Please see Line 321-336.

Reviewer 2 Report

Review Report: The Effect of Green Finance on the Ecological and Environmental Quality of the Yangtze River Economic Belt

Summary

This paper is well written, makes contribution to the science, and can be published after minor revision.

Main Comments and Suggestions

Does the introduction provide sufficient background and include all relevant references?

Yes, the introduction is complete and provide sufficient background. The Authors clearly present the idea, place research hypotheses, highlight the purpose of the study, and briefly describe research methods used. The paper also provides the theoretical framework. The literature review was carried out with great care in each research area analyzed by the Authors.

However, the literature review is based mostly on the Chinese publications. In my opinion, the Authors should also include research provided by researchers from other countries. 

Are all the cited references relevant to the research?

Yes, all the cited references are relevant to the research. 

Is the research design appropriate?

Yes, the research is design appropriate. The Authors support their research approach by the extensive literature framework.  

Are the methods adequately described?

Yes, the Authors describe methods clearly and in details. 

Are the results clearly presented?

Yes, the results are clearly presented. The Authors divided the presentation of the results into sections and additionally illustrated some of the results with tables and figure. Such presentation of the results increases their clarity and facilitates their understanding. 

Are the conclusions supported by the results?

Yes, the conclusions are written properly and are supported by the results. Additionally, the Authors indicate some limitations of their research which illustrates their scientific maturity. Finally, some recommendations were proposed by the Authors to promote green finance development.  

Good luck with your future research!

Author Response

Dear reviewer, thank you very much for your valuable comments and suggestions. According to your proposal, we have revised the paper to get your approval.

Point 1: The literature review is based mostly on the Chinese publications. In my opinion, the Authors should also include research provided by researchers from other countries.

Response 1: We have added some references provided by researchers from other countries to provide a comprehensive and correct summary of national and international research. Please see Line 141-318.
